# Dynamics of Rising Bubbles and Their Impact with Viscoelastic Fluid Interfaces

**DOI:** 10.3390/polym14142948

**Published:** 2022-07-21

**Authors:** Yongjian Zhang, Chenlong Liu, Xiuxing Tang, Xin Dong, Tan He, Heyi Wang, Duyang Zang

**Affiliations:** 1Shaanxi Key Laboratory of Surface Engineering and Remanufacturing, Xi’an University, Xi’an 710065, China; 18009138964@163.com (X.D.); xiaoheas809@163.com (T.H.); why19855590600@163.com (H.W.); 2School of Physical Science and Technology, Northwestern Polytechnic University, Xi’an 710129, China; michael123lcl@mail.nwpu.edu.cn (C.L.); 625064043@mail.nwpu.edu.cn (X.T.); dyzang@nwpu.edu.cn (D.Z.)

**Keywords:** bubble, nanoparticle, surfactant, viscoelasticity, impact dynamics

## Abstract

Bubble dynamics plays a significant role in a wide range of industrial fields, such as food, pharmacy and chemical engineering. The physicochemical properties of complex fluids can greatly affect the speed with which bubbles rise, and the lifetime of bubbles, which in turn can affect the efficiency of food and drug manufacturing and also sewage purification. Therefore, it is of great scientific and practical significance to study the influence mechanism of nanoparticles and surfactants on bubble rising and impact in a complex fluid interface. This paper selects a mixed dispersion liquid of nanoparticles (SiO_2_) and a surfactant (SDS) as the objects of the study, observes in real-time the entire processes of bubbles rising, impact at the gas-liquid interface, and rupture, and analyzes the dynamic mechanism of bubble impact in a complex fluid interface. By analyzing the morphological changes of the rising bubbles, the rising velocity and the lifetime of the bubbles, it is found that the surfactant molecules are distributed in the ultrapure water liquid pool and the liquid film surrounding the bubbles. Such distribution can reduce the viscoelasticity between bubbles and the liquid surface, and lower the surface tension of the liquid, which can reduce the rising velocity of bubbles, delay the drainage process of bubbles on a liquid surface, and enhance the lifetime of bubbles. If the liquid surface is covered with nanoparticles, a reticulate structure will be formed on the bubble liquid film, which can inhibit bubble discharge and prolong bubble lifetime. In addition, the influence of such a reticulate structure on liquid surface tension is limited and its function is far smaller than a surfactant.

## 1. Introduction

Bubble rising is a common physical phenomenon in nature, and the separation of mixtures using bubbles and surfactant is an important method of purification in the chemical industry. Bubbles are also involved in many industrial fields, playing an important role in mineral flotation, food and drug manufacturing, cosmetics and sewage purification [1,2,3,4,5].

The process of bubble rising and impact with the gas-liquid interface is definitely complex. The Van der Waals force between nanoparticles and surfactants has important effects on bubble lifetime and rising velocity. The addition of surfactants or covering nanoparticles at the gas-liquid interface can significantly increase the bubble lifetime, which is mainly because the surfactant and nanoparticles can slow down the speed of bubble liquid film discharge to the liquid pool [6,7,8,9]. When the bubble at the liquid surface is in a stable state, the liquid film above the bubble, the air pressure inside the bubble and the atmospheric pressure are in a relatively stable state. The bubble will rupture when the liquid film keeps discharging liquid to the liquid pool and reaches the threshold value. The following methods can slow down the liquid discharge rate of the bubble to improve the bubble stability: adding organic solvent n-butanol [10,11,12,13,14,15], adding covering nanoparticles [16,17,18], and adjusting the Reynolds number of the liquid phase, which increases the viscosity of the bubble rising [18,19]. In the process of bubble rupture [20,21,22,23,24], the liquid surface slides down to the liquid pool while the liquid surface at the bottom tightens up due to surface tension and eventually forms a jet. If surfactant is added to reduce the surface tension, the jet can be suppressed.

In recent years, the process of bubble rising and impact at complex fluid interfaces has attracted increasing research interest [25,26]. Feng et al. used a double-spring vibration model to accurately describe the vibration of bubbles at the gas-liquid interface [24,25,26,27,28,29,30,31]. Albadawi et al. [32] used the volume of fluid method to make more accurate calculations of bubble rising and impact velocity. Li et al. [33] studied the interaction between bubble floating and the particles, showing that the bubbles did not bounce back significantly when they collide with lower density particles, and the particles were more likely to slip through from around the bubbles. Zawala J. et al. [34] analyzed the relationship between liquid phase viscosity and bubble deformation, finding that the bubble discharge rate was affected by the bubble deformation at the liquid surface, and a more viscous liquid phase would cause the bubble to have a larger deformation, slowing down the discharge rate, thus increasing the bubble lifetime. In order to study the particle attachment to the bubble involving various interactions between the particles and the bubble caused by electricity, Somasundaran et al. [6] analyzed the mechanism of the effect of surfactant molecules on bubble flotation, and concluded that the energy will change when the hydrocarbon chains are adsorbed on the bubble transfer to the gas phase after the addition of surfactant. The adsorption of specific solid particles by the rising bubble, under the action of electrostatic force, provides a theoretical guide for mineral flotation. However, little has been reported on the comparative study of the mechanism of surfactant and nanoparticle effects on bubble flotation.

In this paper, we adopt the method of comparative experiments and select nanoparticles (SiO_2_) and a surfactant (SDS) as the research subjects. We observe in real-time the process of bubbles rising, their impact with the gas-liquid interface, and bubble stabilization on the liquid surface until rupture. The experimental data such as impact amplitude, oscillation trajectory, and bubble lifetime were analyzed to investigate the kinetic mechanism of bubble impact with complex fluid interfaces. We reveal the different ways of bubble motion influenced by the mesh structure composed of nanoparticles and surfactant.

## 2. Methods

### 2.1. Experimental Setup

The experimental device adopts the design of connected vessels, consisting of acrylic pipes, rubber conduits, surface light sources, peristaltic pumps and a high-speed camera (Figure 1).

The heights of the two ends of the acrylic pipe are different. The lower end is in the experimental area, the higher end is in the leveling area. Due to such processes as evaporation that will occur during the experiment, the level of the liquid in the experimental area will drop and separate from the top of the container. In order to avoid the shadow generated by the two parts of the device interfering with the shooting of the bubble pattern, it is necessary to timely replenish the amount of water at the leveling end. We needed to maintain a constant level of liquid, so a peristaltic pump was used to (BT100-2J, Jieheng, Chongqing, China) that was connected to a catheter. The pump injected air into the catheter at a constant speed of 0.030 mL/s. The end of the catheter was connected to a capillary glass tube with an inner diameter of 0.075 mm, and the catheter was extended from the leveling end to the experimental end. Excessive depth will lead to faster bubble rising velocity, so that the bubble could be easily seen and recorded from the shooting area. We fixed the catheter at 10 mm from the liquid surface at the experimental end of the device, producing bubbles of the same size (long axis = 4.372 mm ± 10%) and the size of the initial speed of the floating. The bubbles were recorded with a high-speed camera at 2000 frames/s to record the whole process of bubble rising, impact at the gas-liquid interface, oscillation, and rupture.

### 2.2. Experimental Materials

The experimental liquids in the acrylic tubes were ultrapure water (EPED, China), ultra-pure water covered with nanoparticles (SiO_2_, 150 nm, Wacker Chemie, France) and ultra-pure water with surfactant (SDS, Aladdin Industrial Corporation China). The percentage of nanoparticles containing hydroxyl groups (the higher the percentage of hydroxyl groups, the more hydrophilic the particles) was 35%, 50%, and 75% respectively, and the surfactant was configured as 0.2 cmc, 0.6 cmc, and 0.8 cmc according to the critical micelle concentration.

### 2.3. Experimental Methods

Firstly, ultrapure water was injected into the container leaving a distance of 1~2 mm between the liquid surface and the top of the container. The corresponding nanoparticles or surfactants were added to the experimental area according to the experimental conditions. When adding nanoparticles, the nanoparticles were first dispersed in Aladdin Industrial Corporation China solution, then the dispersed isopropanol solution was added to the experimental area, waiting for more than 30 min until the isopropanol evaporated completely. Water was added to the leveling zone until the surface of the test zone reached the top of the container. The peristaltic pump was turned on to generate bubbles at a constant speed, the whole process of bubbles rising, impacting at the gas-liquid interface, oscillating, and breaking is photographed with a high-speed CCD at 2000 frames/s, and the recorded video was analyzed with Photon Cam Fastanalysis Motion software. Each group of experiments was repeated 20~30 times, and the calculated physical quantities were averaged.

## 3. Experimental Results and Discussion

### 3.1. Bubble Flotation

The bubbles mainly experience three stages: rising, collision, and bursting. Figure 2 shows the collision and rupture process of bubbles at different gas-liquid interfaces. When the bubble impacts the liquid surface, the liquid surface will vibrate violently, and the kinetic energy of the bubble will dissipate with the vibration until the amplitude decreases. Then the liquid film of the bubble top discharges liquid to the pool. The inner pressure of the bubble is the sum of the top liquid pressure and the atmospheric pressure. When the volume of drained liquid reaches the threshold, the bubble bursts. In the rupture stage, the liquid film on the top of the bubble slides to the liquid pool, and the bottom liquid surface tightens and forms a jet under surface tension.

In the rising stage, bubbles are distorted by the viscous resistance of the surrounding liquid. Figure 3 shows the variation of the bubble aspect ratio *K* (*K* = a/b (where a = length and b = height of the bubble) with bubble rising time. It is found that the degree of bubble deformation significantly depends on the surrounding liquid. When the liquid medium is ultrapure water, the *K* value of a bubble is between 1.6 and 2.0 in the rising stage; when there are SiO_2_ particles (35% SiOH, 0.1 g/cm^2^) on the liquid surface, the bubble deformation is aggravated, and the *K* value can reach 2.17; when SDS (0.2 cmc) is added, the bubble deformation is obviously weakened, and the bubble shape is closer to a spherical shape (Figure 2b) [35], and the *K* value is between 1.2 and 1.4.

To test whether the difference in bubble shapes during the rising stage is related to the speed of bubble rise, the time taken for a bubble to rise (the time from bubbles forming to deforming at the liquid surface) was determined experimentally (Figure 4). Compared with the SDS solution, the bubble in the solution coated with nanoparticles has a shorter rising time and flatter ellipse (Figure 3). It can be seen that the bubble is more obviously blocked by the surfactant in the process of its rising, and the bubble rising time increases when the hydrophilicity of nanoparticles decreases. This is mainly related to the distribution of surfactant molecules in the liquid [36]. At the interface of the bubble and the surrounding liquid, SDS hydrophilic particles are absorbed by the surrounding liquid, and hydrophobic particles enter the bubble. In the process of bubble rising, viscoelasticity between the bubble and the liquid reduces the bubble rising velocity. The deformation value *K* is lower than with ultrapure water and covered with nanoparticles on solution surface. The capillary number Ca = μ*v*/σ (where μ is the viscosity of the bulk liquid, *v* is the bubble speed, and σ is the surface tension) involved in bubble rising is 0.01–0.04, indicating the bubble shape is surface tension dominated.

### 3.2. Impact between the Bubble and Gas-Liquid Interface

The kinetic energy of the impact between the bubble and gas-liquid interface, and the dissipation of energy, influence the lifetime of the bubble on the liquid surface. They also reflect the influence mechanism of nanoparticles and surfactants on the rising of bubbles. Thus the experiments analyze the maximum amplitude, vibrating time, and bubble life after the bubble impacts with the surface. By tracing the oscillatory trajectories (Figure 5), it is found that the oscillatory trajectories are similar and the damping effect is weakened by the addition of either nanoparticles or ultrapure water. However, when SDS is added, the bubble settles into stable state after the first impact.

After the bubble impacts the gas-liquid interface, the liquid level oscillates violently. When the solution is ultrapure water, the maximum amplitude of the surface is 3.5 mm. When nanoparticles are added, the maximum amplitude of the liquid surface remains between 2.6 and 3.0 mm, and it is generally unaffected by concentration (Figure 6a). The maximum amplitude of the surface was less than 2.0 mm after adding surfactant. Therefore, adding a surfactant can reduce the velocity of rising bubbles and the kinetic energy of bubbles impacting with the liquid surface.

When a bubble hits the gas-liquid interface, the kinetic energy of the bubble dissipates with the oscillation until the bubble reaches a relatively stable state. Compared with the addition of nanoparticles, the bubbles with surfactant have a shorter oscillation time within 0.1 s (Figure 6b). Although the concentration of nanoparticles has no regular effect on the energy dissipation rate of bubbles at the gas-liquid interface, the time from contact to settling into a steady state is longer than that when the surfactant is added.

### 3.3. Liquid Drainage and Bubble Life of Liquid Film

After entering a stable state, the pressure inside the bubble is the sum of the atmospheric pressure and the pressure of the liquid on the bubble. The liquid film on the top of the bubble continuously drains into the liquid pool, and when the liquid reaches the threshold, the bubble bursts. In the stable state, the direct factor affecting the bubble life is the discharge velocity. Both adding surfactants and covering nanoparticles can significantly prolong the bubble life (Figure 7). The average life of a bubble in ultrapure water is only 0.74 s. The life of bubbles increases with the concentration of surfactant in solution, from 5.667 s to 58 s, and the effect of hydrophobic nanoparticles (SiO_2_, 35% SiOH) is more significant than that of hydrophilic particles (SiO_2_, 75% SiOH).

During the stabilization phase, the surface tension is low. The Gibbs formula can be used to describe this [37]:$$\Gamma =-\frac{c}{RT}{\left(\frac{\partial \sigma }{\partial c}\right)}_{T}$$
where *Г* is surface adsorption, *R* is molar gas constant, *T* is absolute temperature, *c* is the surfactant concentration, *σ* is the surface tension. Generally speaking, the surfactant $\frac{\partial \sigma }{\partial c}$ is negative. Therefore, *Г* is a positive value, which shows that the surface layer concentration is greater than the body phase. There is a positive adsorption, that is, a surface excess, which reduces the surface tension to delay the flow rate of bubbles. This also significantly increases the lifetime of bubbles. The addition of nanoparticles will form a network structure on the gas-liquid interface, which also has the effect of delaying the bubbles from discharging liquid.

### 3.4. Bubble Rupture and Jet Flow

In the process of bubble burst, the liquid film gets thinner and thinner gradually until it cannot withstand the pressure difference between the bubble and the atmosphere. The top of the bubble ruptures and slides down, tightening the bottom surface creating surface tension and generating a jet (Figure 8). In ultrapure water, the instantaneous velocity of the jet can reach 0.85 m/s and the maximum unseparated height is 5.12 mm (no separated droplets on top). However, the jet height decreases with the addition of nanoparticles and surfactants. With the addition of a surfactant, the height of the jet is reduced to 1.92 mm. The contrast pattern change (Figure 2b) clearly shows that the bubble bursts without producing a noticeable jet, but is suppressed into a tiny bulge with a smoother top. In the case of the addition of nanoparticles (0.1 g/cm^2^, 35% SiOH), the bubble burst jet is greatly suppressed at a height of ~1 mm, after which the jet drops back and the vibration spreads around the liquid tank (Figure 2e). The presence of surfactants and nanoparticles at the gas-liquid interface results in an unneglectable interfacial viscoelasticity at the fluid interface [38], which may play a role in suppressing the formation of a jet.

## 4. Conclusions

The different influence mechanisms of nanoparticles and surfactants on bubbles were obtained by analyzing the changing shape of bubbles, their impact with the gas-liquid interface of a complex fluid, bubble oscillation and bubble lifetime. The net structure of nanoparticles on the liquid surface is the main factor affecting bubble deformation, fracture jet and lifetime. However, when the hydrophilic property of the particles is combined with the surface tension and the van der Waals force of the liquid phase, the discharge velocity of liquid film is delayed and the threshold of the thickness of the liquid film is increased significantly. The surfactant dissolves in the liquid pool, which reduces the viscoelasticity of the bubble and the liquid surface, and increases the bubble rising time. The different effects of nanoparticles and surfactants on bubble flotation were analyzed, and the results could prove to be helpful in the improving the efficiency of sewage purification efficiency, and also of food and drug production.

## Figures and Tables

**Figure 1 polymers-14-02948-f001:**
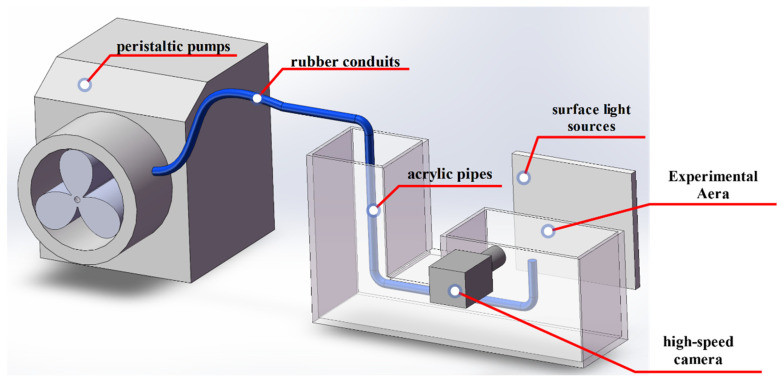
Experimental setup.

**Figure 2 polymers-14-02948-f002:**
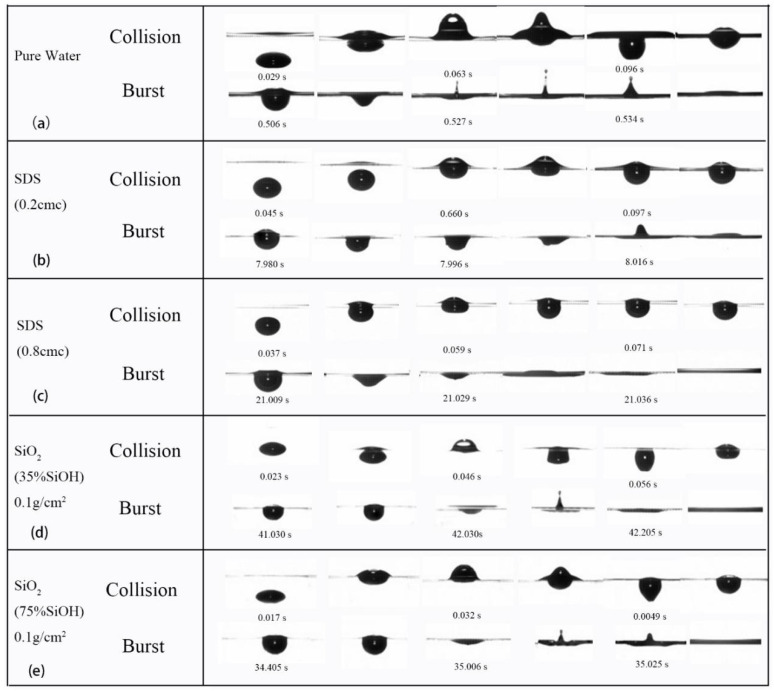
Bubble impact on different gas-liquid interface and its rupture process. (**a**) Ultrapure water condition only; (**b**)the SDS solution with a concentration of 0.2 cmc used in the aqueous phase; (**c**) the SDS solution with a concentration of 0.8 cmc used in the aqueous phase; (**d**) the gas-liquid interface laden with SiO_2_ nanoparticles of 35% SiOH; (**e**) the gas-liquid interface laden with SiO_2_ nanoparticles of 75% SiOH.

**Figure 3 polymers-14-02948-f003:**
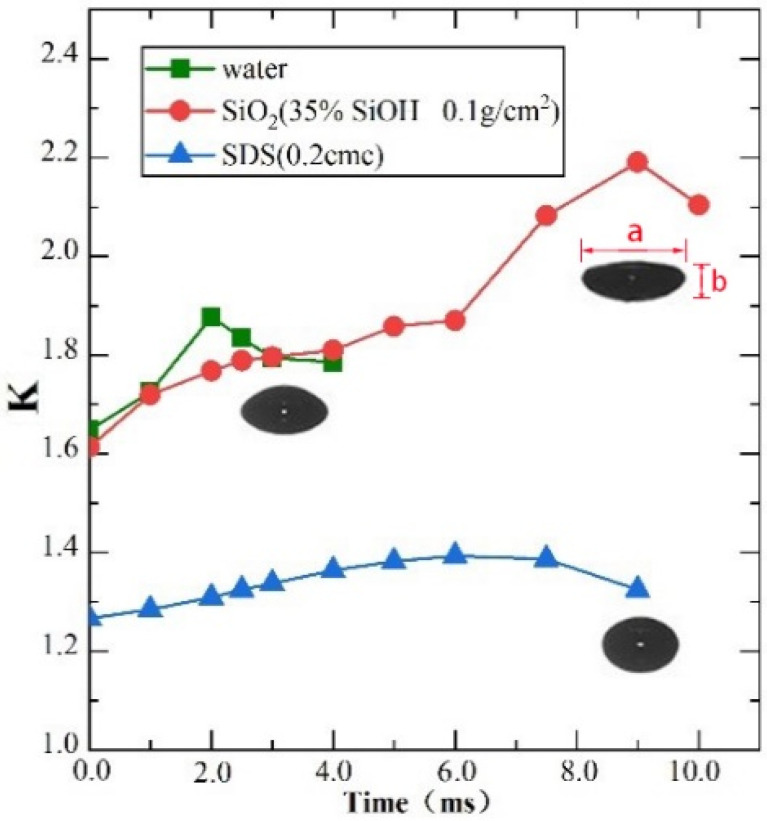
Time evolution of the bubble aspect ratio *K* during its rising, where **a** and **b** are diameters for the major and minor axes of the bubble.

**Figure 4 polymers-14-02948-f004:**
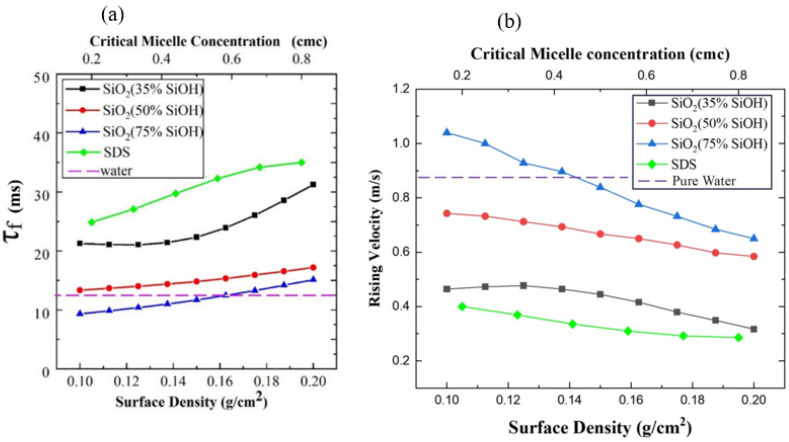
(**a**)Time duration for bubble rising and (**b**) rising velocity versus surface density of particles or SDS concentration.

**Figure 5 polymers-14-02948-f005:**
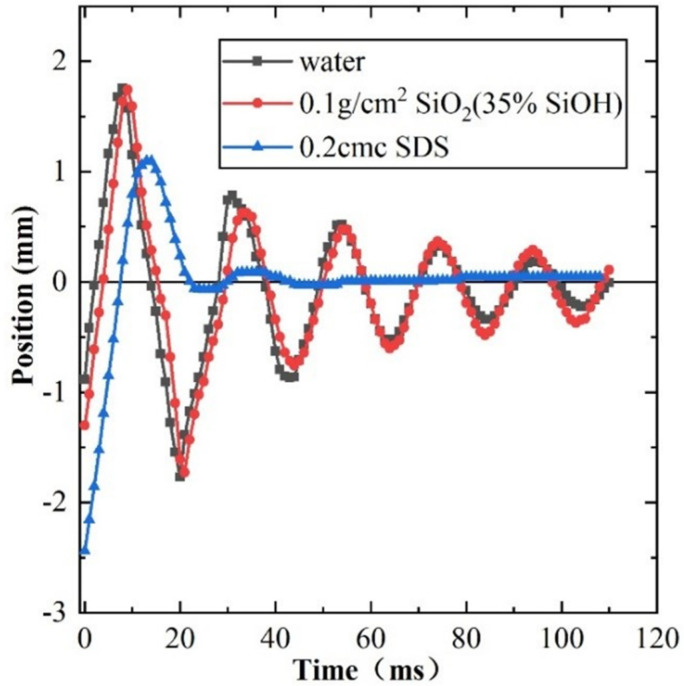
Bubble oscillation at the gas−liquid interface.

**Figure 6 polymers-14-02948-f006:**
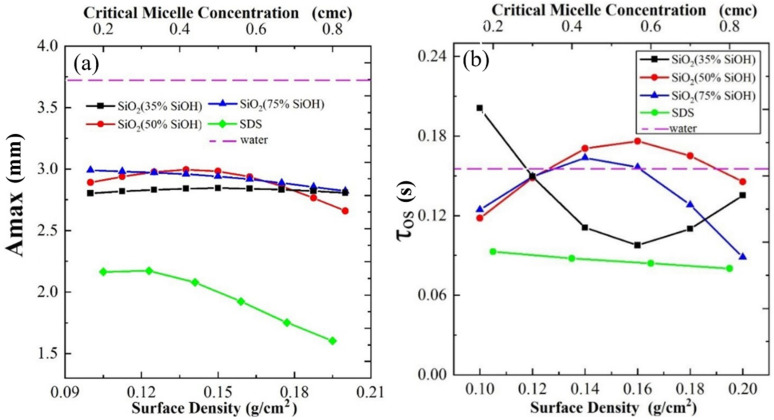
(**a**) The maximum amplitude of waves at the surface when bubbles impact with the gas-liquid interface; (**b**) the oscillation time.

**Figure 7 polymers-14-02948-f007:**
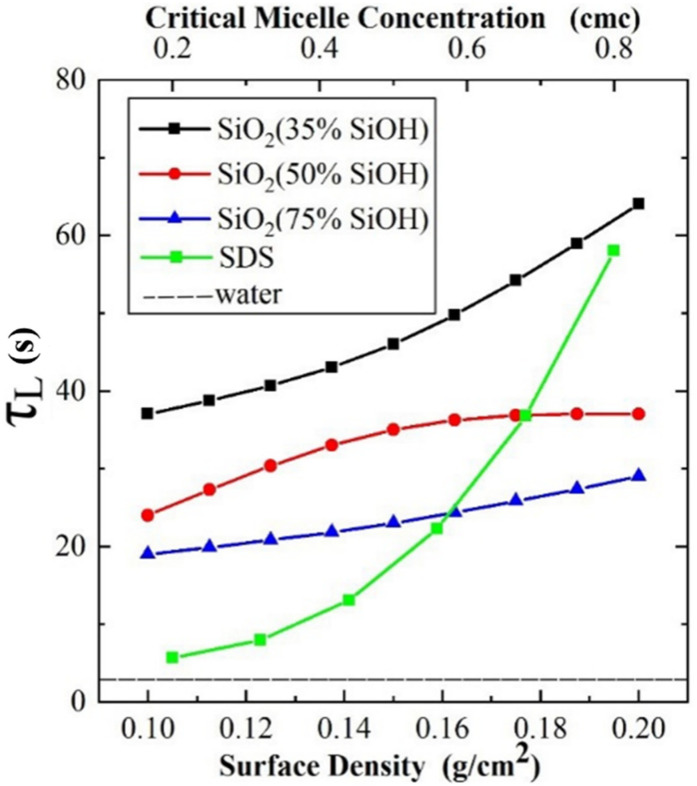
Lifetime of bubbles.

**Figure 8 polymers-14-02948-f008:**
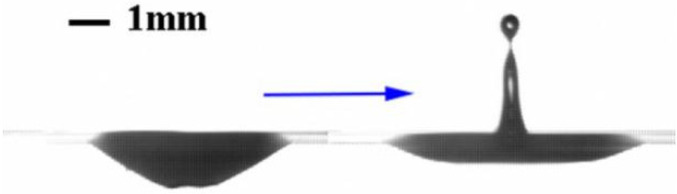
Jetting produced by bubble rupture.

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
