# Peer review of "Dynamics of Rising Bubbles and Their Impact with Viscoelastic Fluid Interfaces"

_polymers, 2022, doi:10.3390/polym14142948_

Round 1

Reviewer 1 Report

Report on the paper by Zhang et al.

It is a very interesting paper on bubble oscillation at the gas-liquid interface as well as the jet formation. Thus, I would like to recommend the publication of the paper after minor changes.

1. The dimension of the vertical axis of Fig. 6 (b) and Fig. 7 should be given in the figures.

Author Response

It is a very interesting paper on bubble oscillation at the gas-liquid interface as well as the jet formation. Thus, I would like to recommend the publication of the paper after minor changes.

Response: We thank the referee for his/her positive comment.

  1. The dimension of the vertical axis of Fig. 6 (b) and Fig. 7 should be given in the figures.

Response: The dimension of the vertical axis of Fig.6(b) and Fig.7 have been added.

Reviewer 2 Report

The authors studied an old problem - the kinetics of rising of bubbles and their impact with the air/water interface. This topic has been somehow forgotten in the recent years, so I congratulate it coming back in the literature. Yet I have some recommendations to the authors:

1. The shape of the bubble depends not only on the complex viscosity of the fluid, but n the size of the bubble. So if they want their study to b complete they should study and bubbles with smaller size - may be in their next works;

2. I advise them t use the capillary number Ca=mu.U/sigma, where mu  is the viscosity of the suspension and U is the speed of the bubble and sigma is the surface tension.

3. Fig 4 could be with the speed of the bubble instead of the rising time.

4. The authors should scrutinize and cite the works of Malysa and Krasowska, who studied the same system many years ago and published number of papers on this. 

After this minor revision I recommend the publication.

Author Response

The authors studied an old problem - the kinetics of rising of bubbles and their impact with the air/water interface. This topic has been somehow forgotten in the recent years, so I congratulate it coming back in the literature. Yet I have some recommendations to the authors:

Response: We thank the referee for his/her positive comment. Bubble rising and impact with the air-water interface is really an old topic, but of importance for both fundamental research and engineering. This topic is one of our main interests in recent years.

  1. The shape of the bubble depends not only on the complex viscosity of the fluid, but n the size of the bubble. So if they want their study to b complete they should study and bubbles with smaller size - may be in their next works;

Response: We agree. We will study bubbles of varied size and clarify the size effect on both bubble shape evolution and the dynamics of interfaces.

  1. I advise them t use the capillary number Ca=mu.U/sigma, where mu  is the viscosity of the suspension and U is the speed of the bubble and sigma is the surface tension.

Response: We thank the referee for the insightful suggestion. The capillary number associated with bubble rising has been calculated to be 0.01-0.04, indicating the bubble shape is surface tension dominated. The discussion has been added in the revised version.

  1. Fig 4 could be with the speed of the bubble instead of the rising time.

Response: We agree. A new figure (Fig.4b) has been added which showed the rising velocity as a function of density.

  1. The authors should scrutinize and cite the works of Malysa and Krasowska, who studied the same system many years ago and published number of papers on this. 

Response: We thank the referee for telling us the interesting papers by Malysa and Krasowska. We have cited several of their papers in the new version.
